# Spotted Wolffish Broodstock Management and Egg Production: Retrospective, Current Status, and Research Priorities

**DOI:** 10.3390/ani11102849

**Published:** 2021-09-29

**Authors:** Nathalie Rose Le François, José Beirão, Joshua Superio, Bernard-Antonin Dupont Cyr, Atle Foss, Sylvie Bolla

**Affiliations:** 1Laboratoire de Physiologie et Aquaculture de la Conservation, Division des Collections Vivantes, de la Conservation et de la Recherche, Biodôme de Montréal/Espace pour la Vie, Montréal, QC H1V 1B3, Canada; 2Faculty of Bioscience and Aquaculture, Nord University, 8049 Bodo, Norway; joshua.superio@nord.no (J.S.); sylvie.bolla@nord.no (S.B.); 3MERINOV, Grande-Rivière, QC G0C 1V0, Canada; Bernard-Antonin.Dupont-Cyr@merinov.ca; 4Akvaplan-Niva Inc., Framsenteret, 9296 Tromsø, Norway; atle.foss@akvaplan.niva.no

**Keywords:** spotted wolffish, broodstock, reproduction, rearing environment, nutrition, sperm handling, cryopreservation, health, welfare

## Abstract

**Simple Summary:**

Spotted wolffish, a cold-water fish species, is a high potential candidate marine fish species for the diversification of the aquaculture sector in Norway, Canada, and Iceland. A review of the state of advancement of all aspects of its reproduction is proposed. Species-specific life-history and reproduction traits are discussed in parallel with relevant information originating from past R&D activities and the current state of knowledge regarding rearing environmental conditions and practices possibly affecting broodstock performances.

**Abstract:**

The first artificially fertilized spotted wolffish (*Anarhichas minor*) eggs hatched in Norway in the mid-1990s as this species was considered by Norwegian authorities to be a top candidate species for cold-water aquaculture in the North Atlantic regions. Previous research conducted in Norway (since 1992) and Canada (since 2000), focused on identifying key biological parameters for spotted wolffish cultivation which led, respectively, to the rapid establishment of a full commercial production line in northern Norway, while Québec (Canada) is witnessing its first privately driven initiative to establish commercial production of spotted wolffish on its territory. The control of reproduction can be viewed as a major requirement to achieve the development of performant strains using genetic selection tools and/or all-year-round production to bring about maximal productivity and synchronization among a given captive population. Although the basic reproduction aspects are more understood and controlled there are still some challenges remaining involving broodstock and upscaling of operations that limit the achievement of a standardized production at the commercial level. Quality of gametes is still considered a major constraint and it can be affected by multiple factors including nutrition, environmental conditions, handling practices, and welfare status. Internal insemination/fertilization and the protracted incubation period are challenging as well as the establishment of a health monitoring program to secure large-scale operations. The profound progress achieved in the control of reproduction, sperm handling, and cryopreservation methods for this species is presented and discussed. In this review, we also go into detail over the full range of up-to-date cultivation practices involving broodstock and identify areas that could benefit from additional research efforts (i.e., broodstock nutrition, health and welfare, scaling-up egg and larval production, genetics, and development of selective breeding programs).

## 1. Spotted Wolffish Farming

Considering the thermal preferences of the spotted wolffish (*Anarhichas minor*), it qualifies as an ideal cold-water aquaculture candidate with a range of attractive characteristics, including high specific growth rates in captivity at very high densities, a high fillet yield (fillets ranging from 0.7–1.1 kg), non-aggressive behavior and few disease problems (See Figure 1). The first breeders of spotted wolffish were gathered in 1992/93 in Norway, and the first artificially fertilized spotted wolffish eggs were hatched in the laboratories of the Norwegian College of Fishery Science, University of Tromsø, in 1994 [1,2]. In the years that followed, a complete production line was established, and a single hatchery (Troms Steinbit AS, Troms County) supplied a single on-growing farm (Tomma Marinfisk AS, Nordland County) with juveniles. In the years 2004 to 2006, approximately 100–120 tons of fish were harvested each year, supplying the Norwegian and Swedish restaurant markets with farmed wolffish at prices ranging from EUR 1–2/kg for fish with harvest sizes ranging between 3 and 5 kg. Prospects were promising until a pump failure resulted in mortality of half the standing stock, as well as all broodfish. The commercial production at Tomma Marinfisk was terminated at this point and spotted wolffish aquaculture activities lay fallow until a new company, Aminor AS, was established in Halsa, Nordland County in 2013. Today, this single commercial farm produces around 50–100 tons of spotted wolffish and is currently up-scaling production to a capacity of 500 tons/year.

In Canada, the unique North American wolffish population in captivity include the original wild-caught fish (1998–2000) and 20 years of cohorts produced during a strategic research program initiated by the Université du Québec à Rimouski (UQAR) (1998–2009) in order to diversify the marine aquaculture sector [3]. The wolffish were identified as a priority species for the initiation of a research program after conducting a multispecies evaluation looking at domestication traits, level of aquaculture knowledge, and market perspectives through a North American context [3,4,5]. The research group from UQAR targeted their efforts at the resolution of the identified bottlenecks limiting the emergence and the development of commercial aquaculture activities facing spotted wolffish cultivation until 2012. It culminated in the realization of a 0.5 MT pilot-scale growth trial targeted at providing potential investors state-of-the-art growth and availability of performance data [6]. The Maurice Lamontagne Institute (Department of Fisheries and Oceans, Canada) took over the research efforts for several years, maintaining the broodstock and conducting various physiological studies of relevance to the fisheries. Aquaculture development efforts are now conducted under the leadership of MERINOV (Grande-Rivière, QC, Canada), based on the presence of private groups of investors committed to taking this species to a commercial level. The resulting large number of domesticated spotted wolffish produced through the years are actually maintained in five different locations across Québec territory, each with its specific environmental conditions and rearing protocols. Two of these locations feature public aquariums offering eco-systemic representations of the Saint Lawrence Estuary (Biodôme de Montréal and Aquarium du Québec) and are more involved in health management of captive populations and conservation activities targeted at this endangered species. Indeed, since May 2001, the spotted wolffish is considered threatened by the Committee on the Status of Endangered Wildlife in Canada (COSEWIC) due to declines in their abundance and biomass [7].

Few reviews on spotted wolffish aquaculture R&D are available [1,2,8]. In the present work, we focused on broodstock management and egg production since these are considered the areas that most limit further expansion of commercial operations.

## 2. Spotted Wolffish Reproduction

Spotted wolffish have a large distribution on both sides of the North Atlantic Ocean and inhabit depths ranging from 25 to 590 m [2,5]. This species is a stenothermal species reported at temperatures from −1 to 7 °C that inhabits stony but also sandy and muddy demersal areas [2,5]. In the wild, the wolffish migrate to colder and deeper waters during the spawning season [9,10] which occurs mostly between July and August [11]. Studies in the closely related Atlantic wolffish, also called common wolffish, *Anarhichas lupus*, indicate this species is solitary in the wild, except during the spawning season when monogamous pairs are formed [12,13] and return to their spawning grounds [14]. It is believed that couples are stable over the years and that they will occupy the same nesting areas every year [15]. Nonetheless, in captivity, the species has been reported to spawn from late summer until the winter [13,16]. Water temperature and photoperiod both have shown to play a role in the spawning time [13,17,18] and differences in spawning timing between the different areas are frequently attributed to temperature [13]. According to Foss et al. [2] and Le François et al. [5], the most suitable temperature for gonad maturation in this species is between 4–6 °C. 

### 2.1. Reproductive Strategy

Wolffish are iteroparous seasonal spawners, with separate sexes (gonochoristic). In the wild, the majority mature at the age of 7–8 years for females, and 8–9 years for males [19] and on average are able to spawn once a year [20] (Figure 1 and Figure 2 below). Wolffish are actually one of the few examples of a marine aquaculture species with demersal eggs, internal fertilization (or internal insemination, see discussion in Section 2.2), that build nests and display parental care apart from the ocellated wolf eel (*Anarrhichthys ocellatus*), a related species [21] for which efforts are being conducted for their cultivation [22]. Contrary to Atlantic wolffish, spotted wolffish natural spawning behavior was never observed in captivity, but similar behavior is attributed [16]. After spawning an egg mass that could comprise as much as 50,000 eggs, the males will wrap themselves around the eggs and create a cluster of eggs. The cluster of eggs is then guarded by the male for the duration of the incubation period until hatching [12,23]. During this period the males cease feeding, whereas the females do so in the later stages of egg maturation [16]. 

### 2.2. Sexual Dimorphism

Sexual dimorphism, apart from the swollen bellies in females in the months before spawning, was never irrefutably reported. Dupont-Cyr et al. [16], reported that in captivity, males grew faster after being freed from the nest- and egg-guarding duties normally carried in the wild. This phenomenon was observed in both species of wolffish of interest for aquaculture. Male spotted wolffish reached a 15–20% higher somatic weight than females at the end of the growth trial. The production of all-male stocks should, according to these authors, be considered. In this context, the findings of Maltais et al. [24] that used purified vitellogenin to develop an indirect competitive ELISA for the determination of gender and sexual maturity of females is of interest, although sex determination would need to be applied earlier in development. One study in the wild mentioned that males grew more in length than females post-maturation [20].

In captivity, it is reported that fish handlers can usually recognize the males from the females, due to more elongated heads for the latter vs. a larger and rounder head for the former. Furthermore, males may display increased activity during the spawning season. However, no study has yet shown quantitative or qualitative evidence of these differences in morphometry or activity levels between the two sexes.

### 2.3. Internal Fertilization/Insemination

Some discussion still remains as to whether this species displays external fertilization, internal insemination (copulation and delivery of the sperm in the female ovary) and subsequent external fertilization or both internal insemination and fertilization. Kime and Tveiten [25] present some arguments in favor of external fertilization and insemination, such as the absence of an evident anatomical adaptation to deliver the sperm in the female genital pore. Nonetheless, all the remaining literature [2,8,26] points toward probable internal insemination and subsequent internal fertilization as is the case for the Atlantic wolffish [27] (see video footage. Available online: https://www.youtube.com/watch?v=WLu_pB4BYjM (accessed on 24 September 2021)). In the Atlantic wolffish, Pavlov [27] describes that although the eggs could be fertilized internally and go through a cortical reaction, they need to be released into the seawater before the cleavage begins in order to develop normally. Furthermore, the presence of papilla in the male urogenital pore that becomes more protuberant during the spawning season has been described [28]. Several marine sculpins present a strategy with internal insemination but external fertilization, in a mechanism called internal gamete association [29,30,31]. However, some other sculpin species also inject the sperm in the viscous ovarian fluid immediately after the eggs spawning are involved in the gelatinous ovarian fluid, since the sperm is poorly motile in seawater [32], and a similar mechanism is suggested by Kime and Tveiten for spotted wolffish [25]. Unfertilized egg masses released by unstripped females are also sometimes found in rearing tanks. The exact normal spawning behavior and fertilization in spotted wolffish will remain a source of debate for several years because of the difficulty of observing natural behavior in the wild and the failure to induce natural spawning behavior in captivity.

Challenges with broodstock housing and providing them with the best environment have led not just to the inability to obtain natural spawning but also affect the gamete quality. Simultaneously, suboptimal fertilization techniques and egg incubation procedures have resulted in some level of unpredictability of juvenile production essential for the feasibility of the aquaculture industry for this species.

## 3. Broodstock Capture and Adaptation

Farmed spotted wolffish mature in captivity and the eggs needed for commercial production may easily be obtained from a standing broodstock of first- or second-generation farmed broodfish. However, potential problems may arise, such as variable gamete quality in farmed origin broodstock or lack of control over the genetic background of parental crossings, that could eventually cause inbreeding issues. For these reasons, there is still a need to supply the broodstock with wild-caught fish to sustain a healthy gene pool in the broodstock population or practice controlled-breeding of previously genotyped individuals associated with the use of a pairing matrix based on a relatedness coefficient [Le François N.R. pers. comm.].

There is no targeted commercial fishery for the species and landings mainly consist of by-catches in other fisheries and long-line fishing. In the last few years, landings of wolffish (spotted and Atlantic combined) in the North-East Atlantic have been relatively stable at around 10,000 MT/year (Dept. of Industry Statistics, Statistics Norway). Only on a few occasions have wild fish been captured to be kept for broodstock purposes both in Norway and Canada, and there is little documented experience apart from a couple of recent attempts. In Norway, 30 individuals were caught by long-line in October 2016 at a depth of approximately 250 m, kept in holding tanks with running water on the fishing vessel and transported to the research facilities of Akvaplan-niva (www.akvaplan.niva.no, (accessed on 24 September 2021)), where they were quarantined in an 8 m^3^ tank. Within the first few days, a mortality rate of 15–20% was observed. The remaining fish received an intramuscular injection with antibiotics, but during the next few months several fish died, and none accepted formulated feed in this period. As of today (2021), only two of these fish are still alive, but this first catch of wild broodstock since the mid-90s has provided us with a lot of information on procedures for both capture and adaptation of spotted wolffish to a culture environment. 

In Québec, Canada, the establishment of a captive longlined wild-caught broodstock population, originating mostly from Baugé Bank near Anticosti Island in the Gulf of Saint Lawrence (Québec, Canada), was very successful in providing the means to initiate numerous applied studies over the years and produced an entirely genotyped domesticated population. They were held at the CAMGR and Maurice Lamontagne facilities for numerous years before being shared with other stakeholders and research institutions in 2018 [8]. The findings presented below were implemented following subsequent collections of wild broodstock by Norwegian and Canadian groups (for research and commercial interests).

### 3.1. Capture and Transport

Ideally, the wild fish should be caught during summer/early autumn. At this point, the gonad build-up is in progress, and ripe males and females may be stripped for sperm and eggs already at the first season in captivity. From experience, we know that they may not necessarily spawn the following season, but for culture purposes a fresh start is important. Hauling of the longlines should be performed in a steady motion to prevent unnecessary stress. Spotted wolffish lack a swim bladder, so the pressure difference is not as big a concern as it would have been with other species. When landing the fish, the branch-line should be cut, leaving the hook to be removed later. After transport in oxygenated transport tanks on the ship and until reaching the quarantine facility, the fish should preferably be anesthetized and injection with antibiotics should be provided as soon as possible. At the same time, it may be possible to determine the sex of individual fish and tag them with a color code. For large wolffish, it is common to use livestock ear-tags on or near the dorsal fin. Within the next few days, it is recommended to perform a bath treatment with 1:3000 formaldehyde (37%) to rid the fish of any potential external parasites. It is recommended to repeat this treatment after approximately 1 week to prevent newly hatched cysts or eggs from developing further, as a proportion often survive the first formaldehyde treatment.

### 3.2. Adaptation

The main challenge in adapting wolffish to culture conditions is the transfer from eating wild prey to feeding on formulated feed. From experience, these fish are very reluctant to feed on pellets introduced into the tank, and a lot of hours are needed trying to lure them into eating pellets with different forms and flavor at various times of the day. In our experience, the best way to introduce this new diet is to train the fish, and this means that the caretakers must show them that formulated feed is, in fact, appetent. By mixing wild fish with farmed fish that are already familiarized with pellets, this transition occurs more rapidly than in a group consisting only of wild fish. It is, however, important that the precautions described above are completed before introducing the wild fish into the population of farmed fish, to prevent the spreading of various pathogens. In Canada, wild-caught fish were never weaned on a commercial feed but fed various prey fish and invertebrates with vitamin caplets inserted in the abdominal cavities of small fish. Wild fish were always kept separated from the domesticated fish, the goal being to apply more efforts to the establishment of an entirely domesticated broodstock population.

## 4. Culture Environment and Welfare

In most fish species, a suitable culture environment and ensuring the welfare of the captive fish results in good growth performances and development that leads to successful maturation. On the contrary, poor welfare exerts significant negative effects on growth, behavior, disease tolerance, and sexual maturation. Therefore, the water quality, tank size, stocking density, and feeding should be optimal and well-suited to the cultivated species [33]. Spotted wolffish broodstock are kept in land-based facilities, and during the spawning season, small tanks (e.g., 2 m × 2 m × 0.4 m, [33]) are preferred to facilitate individual fish monitoring, especially during the spawning events. Spotted wolffish land-based rearing systems, like other land-based aquaculture systems, are prone to fluctuations in various parameters due to low water volume and high stocking density. Light intensity is generally kept low as wolffish are accustomed to low light levels in nature. Studies report intensities varying from 0.2–2 µmol m^−2^ s^−1^ [34,35,36]. The effects of the culture environment on broodstock performance and welfare, except for temperature, have seldom been studied in spotted wolffish. Dupont-Cyr et al. [16] studied the growth of adult spotted wolffish and evaluated steroids synthesis and gamete production under different photoperiodic treatments [13]. Accordingly, most of the knowledge presented in this section refers to studies conducted with juveniles, the closely related Atlantic wolffish, and practical approaches adopted in the different institutions housing spotted wolffish broodstock.

### 4.1. Water Quality

Water quality remains a crucial factor involved in fish welfare. Although some fish species can survive in poor water quality, rapid changes in quality parameters can compromise fish welfare and result in mortalities. The measured variables include temperature, dissolved oxygen, dissolved carbon dioxide, and pH, among others [37]. Therefore, it is critical to determine the water quality standards which govern the broodstock production indices (e.g., maturation, egg production, sperm production), fish welfare, and health conditions required by a species to achieve maximum output potential [38]. 

#### 4.1.1. Temperature

In general, fish exhibit an optimum temperature range for growth and maturation. In the spotted wolffish, the optimum temperature for growth is around 8–12 °C [39,40,41]. Because of the central role of temperature in the environmental control of reproduction of spotted wolffish, the temperature is discussed in more detail in Section 6.1. 

#### 4.1.2. Salinity

Although the wolffish are stenohaline marine cold-water species, thriving in depths with minor fluxes in temperature and salinity, several studies have reported that both Atlantic wolffish and spotted wolffish possess an outstanding osmoregulatory mechanism which enabled farmers to rear them in a wider range of salinities (euryhaline) [34,42,43]. Specifically, the Atlantic wolffish juveniles can tolerate a salinity range of 7–35 Practical Salinity Unit (PSU) and achieve an effective homeostatic control even after extended exposure to various salinities, resulting in a slight growth advantage for the fish reared at 14 PSU [42]. Meanwhile, spotted wolffish reared in a salinity range of 12–34 PSU at a fixed temperature of 8 °C showed no significant effect on growth performance [34]. Better growth performance in juveniles was recorded in 25–34 PSU at 10 °C while the opposite effect was observed at a lower temperature [43]. However, no studies exist on the effect of salinity on broodstock performance and currently, the spotted wolffish breeders are reared in a salinity range of 26–34 PSU [13,25] with no detrimental effects reported. 

#### 4.1.3. Dissolved Oxygen, Carbon Dioxide, and Ammonia

Dissolved oxygen (DO) is considered a limiting factor of survival in every aquaculture system since oxygen facilitates all cellular functions. Spotted wolffish are not a very active species, and they spend most of their time dwelling on the tank floor close to each other, and even laying on top of each other. If the biomass is high, this grouping of fish could even result in low oxygen levels in pockets of water surrounding the fish, which is important to consider when measuring oxygen levels in a wolffish rearing tank. Providing good mixing of tanks and air or oxygen supplementation renders poor circulation events infrequent in well-designed culture environments. In spotted wolffish juveniles, both very high (14.5 mg·L^−1^ and low (6.0 mg·L^−1^) DO concentration resulted in a comparable growth after a period of adaptation [44]. Although DO levels ≤ 50% saturation resulted in stunted growth in spotted wolffish [45], further reduction in DO saturation to <21% resulted in a high mortality rate which reached up to 50% within four days of exposure [46]. Meanwhile, it is recommended that broodstock tanks have a 100% oxygen saturation and the effluent concentration at the outlet drain should indicate a >60–70% oxygen saturation level [15]. Studies on spotted wolffish broodstock usually aim to maintain a DO saturation level of >90% in the rearing tanks [13,16,25]. Feeding events often result in the lowering of DO values but, in most cases, they are punctual and of short duration. 

Dissolved carbon dioxide (CO_2_) concentration can increase especially in recirculating aquaculture systems (RAS) and this provides several implications in the physiological functions of the cultured organism. In the spotted wolffish juveniles, varying CO_2_ levels (1.1–59.4 mg·L^−1^) at constant temperature (6 °C) and salinity (33‰ PSU) did not affect feed conversion efficiency but overall growth rates were reduced at the highest concentration [47]. Moreover, nephrocalcinosis (calcareous deposition in kidneys) was observed in all the treatment groups but was more pronounced in the medium to high levels of CO_2_ [47]. In salmonids aquaculture, the reported CO_2_ level limit is 10–20 mg·L^−1^ [48,49]. Nevertheless, Le François and Archer [15] suggested keeping a CO_2_ level of 1.1 mg·L^−1^ in the broodstock tanks of spotted wolffish to avoid the detrimental effects of the toxic compound through blood acidification and oxygen uptake reduction [50].

Nitrogenous compounds (e.g., total ammonia) are byproducts of fish metabolism excreted into the water. In the spotted wolffish juveniles, growth rates were significantly affected by exposure to different unionized ammonia concentrations at a constant temperature, salinity, and pH. Control (0.0006 mg·L^−1^) to low (0.13 mg·L^−1^) ammonia concentration groups performed better in comparison to medium (0.25 mg·L^−1^) and high (0.39 mg·L^−1^) concentrations. Although an adaptive capability of the cultured fish to the existing ammonia levels in the water over time was reported [51], nevertheless it is suggested to keep the ammonia level of the broodstock tanks as low as possible (≈0.0006 mg·L^−1^) to eliminate any risks [15].

### 4.2. Stocking Density, Handling, and Rearing Environment

Stocking density remains a primary factor that contributes to fish welfare. Exceedingly high fish stocking density frequently leads to injuries (external and internal), aggression, behavioral changes, and disease outbreaks [52]. Nonetheless, the tolerance to the high rearing density of the spotted wolffish is a major reason for its popularity as an emerging species for aquaculture diversification [2,3,4,8]. For flatfish and wolffish, rearing densities are commonly expressed in kg·m^−2^ instead of kg·m^−3^ since they occupy the tank bottoms and not the water column. In spotted wolffish juveniles, optimal rearing densities vary between <40 kg·m^−2^ for smaller fish and ≥40 kg·m^−2^ for larger individuals according to Tremblay-Bourgeois et al. [53]. Meanwhile, adult spotted wolffish (1.5–3 kg) are regularly reared at a stocking density of >70 kg·m^−2^ (210 kg·m^−3^). 

Another important welfare issue is the immediate environment or surroundings in which the fish are kept. Even though in the wild, spotted wolffish live and spawn in nests in shelters, the use of shelters was never tested in adults. However, experiments conducted with juveniles indicated that they preferred habitats with shelters and that this did not trigger aggressive or territorial behaviors [54]. Nonetheless, no welfare or stress parameters were evaluated, and thus it is difficult to speculate if such environments could affect reproductive performance. In addition, rearrangement of the volume/area in tanks with the installation of shelves might prove very promising for the industry once land-based commercial production levels increase drastically either for on-growing purposes or broodstock holding.

Stress response in captivity conditions has been briefly studied in spotted wolffish juveniles [53,55,56] but was never evaluated in mature individuals. The slow and somewhat weak cortisol response and low plasma glucose levels found in juveniles may relate to their sedentary lifestyle. The stress response is characterized by a passive–reactive coping style, which is considered adaptive for fish farming activities [55]. Wolffish broodstock are considered long-lived (6–9 years is considered their peak of reproduction) and the impacts of gradual aging and of cumulative acute and/or chronic stress under commercial rearing environments on their health have not been fully addressed or studied, thus the precautionary principle is recommended.

### 4.3. Health and Diseases

At this stage of development, data on diseases and parasites that are of concern to spotted wolffish aquaculture are rather scarce. In accordance with reports emanating from the Norwegian spotted wolffish farming activities, early investigations, and surveillance programs, this species can be considered robust with a low mortality rate, displaying low susceptibility to disease in comparison to other novel farmed marine fish species [8]. However, the forthcoming development of wolffish aquaculture in Europe and North America and the expected level of aquaculture production targeted by governmental agencies and the private sector will likely stimulate the emergence of clinical health expertise to cope with inevitable health challenges. Wolffish health issues, as with most species, are linked to parasitic infestations, bacterial or viral diseases, and given our relatively limited comprehension of the spotted wolffish response to stress or welfare or specific nutritional or environmental needs, chronic unoptimized or detrimental rearing practices are most probably involved in exacerbation of vulnerability to infestation and disease [8]. 

Infections with protozoan parasites such as *Ichthyobodo* sp., *Trichodina* sp., and *Pleistophora* sp. are the most common and severe health issues in farmed spotted wolffish [57,58] and have caused some problems. As an example, in spring 2020, *Trichodina* sp. bloom was responsible for the massive loss of newly hatched wolffish in Canada, and in 2007–2009 it was also reported in juveniles [6]. Ectoparasitic monogeneans of the genus *Gyrodactylus* occasionally have also caused skin lesions, but the only *Gyrodactylus* sp. described from spotted wolffish so far is *Gyrodactylus anarhichatis* [59]. The latter reported in great detail a *Gyrodactylus* infestation in Icelandic farmed spotted wolffish, whereas, Chabot et al. [6] reported summer infestations in rapidly growing juveniles. Both *Trichodina* and *Gyrodactylus* have been treated with formalin bath of the infected fish. These events most likely can be controlled better through proper and regular screening and adoption of filtration and UV treatment, at least for the early juvenile stages. Rearing at intermediate salinities also could exert some degree of control.

Nonetheless, the greatest mortality observed, and the only bacterial disease reported in spotted wolffish prior to 2020, involved atypical *Aeromonas salmonicida* (atypical furunculosis) infections that occurred under stressful rearing conditions. Vaccines have been developed using bacterial strains isolated from spotted wolffish [58,60]. However, wolffish may be affected by other bacteria and viruses. Recent necropsies conducted on the captive domestic broodstock population of the research facilities of Merinov inc. (Gaspé, QC, Canada) detected an important occurrence of *Vagococcus salmoninarum* [Farley, J. MDV unpublished data]. Although only reported in juveniles, spotted wolffish were found to be susceptible to infection with a nodavirus isolated from Atlantic halibut (AHNV) by bath-challenge with accumulated mortality reaching up to 52%. Clinical signs showed similarity to viral encephalopathy and retinopathy (VER), and the causative agent can retain its virulence even after 16 weeks post-challenge [61]. Another virus of concern is the birnavirus which is the causative agent of infectious pancreatic necrosis (IPN). Experimental viral bath challenge resulted in low mortality but persistent viral infection detectable up to 4 months later [62]. 

Additionally, chronic health diseases such as xanthomatosis and nephrocalcinosis have been reported in aquarium populations by Béland et al. [63], and nephrocalcinosis (Figure 3 below) is commonly observed in Norwegian broodstock [Beirão J., pers. comm.]. Béland et al. [63] attributed these problems to unbalanced diets, especially lipid and mineral content, but recognized that more information is needed to understand the actual causes. Meanwhile Chabot et al. [6] reported hepatic lipidosis in on-growing and young adult fish fed a commercial feed with 15–18% lipid content. 

## 5. Nutrition

As is the case with most farmed marine fish species, the overall picture of nutritional requirements of spotted wolffish breeders is fragmented. No detailed studies or feed trials on the effect of broodstock nutrition on egg quality or health have been carried out in controlled conditions. Because spotted wolffish are an iteroparous group synchronous spawner which present highly variable egg quality, great consideration should be given to the broodstock diet to provide the necessary nutrients in adequate amounts to sustain a high egg quality and fecundity over the productive life of the brood fish. The challenges of wolffish broodstock management can be tied to their longevity but most certainly to the absence of adapted diets and, consequently, unmet nutritional needs during the critical and repeated period that is sexual/gonadal maturation. Furthermore, an optimized diet becomes crucial when the life cycle is closed and broodfish are recruited from production. In this regard, little effort has been made for the development of specific diets for this species, in turn limiting further development due to poor gamete quality. Although different broodstock holders have tested different commercial diets, there is not a specific diet formulation available for wolffish breeders. Feeds offered are typically based on fishmeal as the main protein source and fish oil with limited levels of plant ingredients. In Canada, the recent availability of domestic broodstock (born in captivity) in large numbers now offers the perfect conditions to conduct controlled studies on the effect of diet composition on gamete quality [8]. Preliminary work initiated by Desrosiers et al. [64], produced a baseline study involving the development of catabolic and energy metabolism capacities during embryonic development to help identify factors associated with egg quality variability amongst spotted wolffish broodstock congeners. 

Wolffish are a demersal carnivorous species with a range of prey that belong mostly to lower trophic levels. With their powerful jaw and palatine and protruding teeth, wolffish break the shells of sea urchins, crustaceans, various kinds of shellfish, and gastropods. Occasionally, other fish species were observed in stomach contents [10,65]. Nonetheless, they seem to be mostly opportunistic with the stomach contents varying between regions and seasons [66,67]. The natural diet provides a high content of easily digestible protein associated with low lipid content together with shell fragments. In addition, in the wild wolffish stop feeding during the spawning season, coinciding with their annual teeth replacement [65], something that is absent in farmed fish [16]. Fasting occurs first in females, about one month before ovulation [68], and then in males during nest guarding [12]. Thus, specific broodstock diets adapted for this species should be offered in the year preceding expected sexual maturation.

### 5.1. Protein Requirement

Protein requirements are generally high in carnivorous marine fish. Still, all farmed carnivorous marine species have their amino acid requirement covered by a fish meal which is the main protein source in the broodstock diet. Despite the similarity in amino acid composition between fish meal [69] and the natural diet of wolffish, the palatability and digestibility of the fish protein in this species are not documented. Studies indicate that both the spotted and Atlantic wolffish showed good growth rates at protein levels above 50% [2,70]. Nonetheless, a study in the Atlantic wolffish tested protein content from 35 to 59% and showed that protein contents of 54 and 59% gave a higher growth rate when using fish meal as the main protein source [71]. 

### 5.2. Lipid Requirements

Lipid requirements for spotted wolffish are scarcely documented. However, the natural diet of wolffish, such as sea urchins and mollusks, is low in lipids, which indicates that their lipid requirements should also be low. The total lipid content in wild sea urchin (*Strongylocentrotus droebachiensis*) gonads is around 6% [72]. In other invertebrates, such as shellfish, the total lipid content is even lower, in the range of 2 to 3% [73]. These invertebrates present amounts of n-3 HUFA in the range of 95 to 510 mg/100 g meat, with an EPA content generally higher than DHA, dependent on the species. The DHA/EPA ratio in farmed spotted wolffish eggs is 1.18–1.21 [73]; Beirão et al. in preparation], which is one of the lowest ratios for cold-water marine species and a possible consequence of the distribution in the prey. ARA content is low, as expected in a cold environment. Current diets based on fish oil would cover most of the fatty acid requirements even if some imbalance can remain.

### 5.3. Carbohydrates

Carbohydrate digestibility is generally limited in marine species since the carbohydrates are only present as traces in the natural diet. However, the natural diet of the spotted wolffish includes invertebrates that present some levels of glycogen, mostly <10%. A 10% of soluble carbohydrate is however necessary for the extrusion process in the production of dry pellets. Unfortunately, the digestibility of carbohydrates in dry feed is not documented for this species.

## 6. Control of Reproduction

In intensive aquaculture, fish reproduction can be controlled by environmental manipulations (e.g., photoperiod, water temperature, spawning substrate). The lack of a natural environment, coupled with the inevitable rearing stress, results in the various reproductive dysfunctions that are exhibited by numerous fish species of commercial importance [74]. In males, these reproductive dysfunctions include reduced milt production or milt quality [75]. Similar dysfunction is attributed to spotted wolffish males that display an absence of spawning behavior accompanied by a small supply of low-concentration sperm [25]. In females, there are three prevailing captivity-induced reproductive dysfunctions [75]. The first complication is the failure to undergo vitellogenesis. The second (most common) kind of reproductive problem in females is the absence of oocyte maturation (OM), even if late-vitellogenic oocytes were developed [76,77]. The third type of dysfunction, which occurs in spotted wolffish, is the failure of fish to spawn. Different species that exhibit this problem undergo normal vitellogenesis, maturation, and ovulation but the ovulated eggs are not released. These eggs remain in the abdominal cavity and are reabsorbed [78] or can be released at some point after ovulation even without spawning behavior [79]. In captivity spotted wolffish females often fail to release the eggs and become egg bound or release overripe eggs. 

### 6.1. Environmental Manipulation

The failure of captive fish to successfully initiate and conclude gonadal maturation, ovulation/spermiation, and spawning is due to the deficiency of gonadotropins: follicle stimulating hormone (FSH) and luteinizing hormone (LH), secreted from the pituitary gland necessary for gametogenesis and positive regulation of reproduction, all under the influence of external factors (e.g., photoperiod, temperature, food supplies, pheromones) that may act on the hormonal cascades of the axis brain–pituitary–gonads [80,81]. Manipulations of various environmental parameters can often improve spawning consistency. Temperature and/or photoperiod are environmental factors that can be modulated to influence/synchronize endogenous rhythms that offer a best-fitted situation in commercial fish breeding facilities by improving spawning consistency [82,83,84,85] to reach sustainable and profitable production levels.

#### 6.1.1. Thermal Control of Spawning

The temperature has a major influence on many aspects of fish reproduction, including gamete development, maturation, ovulation, spermiation, and spawning [86]. In wolffish, egg quality is frequently low when the individuals are kept at temperatures above 8 °C in the months leading up to spawning. Nonetheless, the effects of temperature on wolffish performance have only been studied in Atlantic wolffish [18,87,88,89]. In the Atlantic wolffish, different rearing temperatures (4, 8, and 12 °C) showed notable effects on its reproductive physiology. Exposure to a higher temperature (i.e., 12 °C) during vitellogenesis resulted in a lower level of maturation-inducing steroid (MIS) 17,20 β-P compared to lower rearing temperatures, suggesting that synthesis and/or metabolism of steroid hormones are influenced by temperature [88]. This effect of temperature was not observed in male fish, the steroid plasma concentrations of which were low at all times during the course of the study [88]. Further studies on the effects of temperature treatment in the temporal variations in plasma testosterone (T) and oestradiol-17 β (E_2_) during ovarian development showed that peaks of both hormones exposed to 8 and 12 °C were delayed by a month in comparison to fish kept at 4 °C. Correspondingly, shifts in the timing of ovulation were observed [89]. A delay of four to five weeks in maturation and ovulation in fish held at 8 and 12 °C compared to 4 °C was also observed by Tveiten and Johnsen [87]. Moreover, fish held at 8 °C produced a significantly high number and larger eggs. Nevertheless, significantly higher normally cleaved eggs and egg survival was reported at 4 °C [18]. These results indicate that temperature affects sexual steroids cycle, ovarian growth, maturation, ovulation, and egg quality in Atlantic wolffish. Overall, these findings might also be applicable to the spotted wolffish due to their similar biology and behavior. Interestingly, Lamarre et al. [90] observed that the environmental temperature profiles during the ovarian maturation period in the area from which egg masses of Atlantic wolffish were collected impact the quality of the egg batches and subsequent hatchlings. Thus, spotted wolffish breeders can be kept at a temperature of 9–10 °C outside the spawning season but during final maturation (up to three months before spawning corresponding with the vitellogenesis), fish should be kept at temperatures below 6 °C [5,8,25,91].

#### 6.1.2. Photoperiodic Control of Spawning

Spotted wolffish respond well to photoperiod manipulation. Earlier results from Norway involving a commercial producer [2] reported that a complete out-of-shift (6-month cycle) was achieved by rearing fish under two consecutive 9-month cycles (1999–2001). When compared to a control broodstock (natural photoperiod), spawning after 18 months in the manipulated group resulted in a lower percentage of fertilized eggs, a lower relative fecundity, and smaller eggs, while no difference in egg survival (% fertilized) was observed. Additionally, at the Nord University, two broodstock were kept, one with a normal photoperiod and another with a reversed photoperiod, in order to obtain spawns in early summer. Unpublished observations at Nord indicate that if fish are exposed to the reverse photoperiod before their first maturation, gamete quality is not affected by this reversed photoperiod. Nonetheless, adult fish that are moved into a new photoperiod will have their gamete quality affected, especially the females that in some cases after 3 years were still maturing in the natural spawning season. Dupont-Cyr et al. [13] provided a more detailed approach that includes monthly plasma and steroid concentration profiles, oocyte diameter growth, and milt production, which offers practical information in order to implement the use of photoperiodic manipulation for the control of sexual maturation of Atlantic wolffish repeated spawners and first-time spawners of spotted wolffish. The experiments lasted 23 months (February 2006 to December 2007) under a compressed photoperiod (8 months) and results strongly suggest that photoperiodic treatment successfully induced a 6-month temporal shift in sex steroid profiles. It was coupled to a successful altered ovulation/final maturation that occurred in comparison to control groups held under a simulated natural photoperiod. Control of reproduction using a photoperiod also enables captive broodstock population to be exposed to incoming water of a temperature closer to the optimal temperature normally observed during natural post-ovulation stages and allowed the avoidance of warm summer water temperatures that precede the natural spawning season in the areas designated for wolffish aquaculture (typically in the range of 10–20 °C during the summer months). Photoperiodic control of actual populations held at Merinov facilities in Grande-Rivière (QC, Canada) and the commercial producer Aminor AS (Norway) is gradually being implemented to support the needs for eggs and juveniles for year-round on-growing activities. Preliminary results indicate a successful shift of the spawning season but also to some degree, spawning hour.

### 6.2. Endocrinology and Hormonal Therapies

In some species, hormonal therapies remain the best option for reliable control of reproduction. However, only a few attempts have been made in spotted wolffish to use hormonal treatments for improvement of the reproductive performance, and no published paper has reported an extensive study on the effect of hormonal therapies on the reproduction of spotted wolffish, especially in females. Both at Nord University (Norway) and at the facilities of MLI (Maurice Lamontagne Institute, Department of Fisheries and Oceans, Mont-Joli, QC, Canada) small trials were conducted to hormonally induce spawning in egg bounded females, using commercially available gonadotropin-releasing hormone agonists (GnRHa). At Nord University, the best results were obtained with two doses of 0.5 mL·kg^−1^ given one week apart [unpublished data]. At IML, spawning of females injected with GnRHa was accelerated by the order of 26.6 days in comparison to uninjected females at the same stage of ovarian maturation. No effect on egg diameter could be observed [Savoie et al., unpublished results]. In males, intramuscular injection of 300 µg·kg^−1^ GnRHa in combination with a dopamine agonist (Pimozide) resulted in an increased level of testosterone and 11-ketosterone (KT) in the plasma two weeks post-treatment. The sperm motility and density also increased in response to elevated plasma levels of steroidal sex hormones [Foss et al., unpublished results]. Simultaneously, some publications have also reported the effect of steroids on male breeders while only preliminary results were reported in the females [92]. Steroids have been described to have an influence on the sperm production and seminal fluid physiology of male wolffish. The administration of cortisol reduced the pH of seminal fluid and the number of milting males. On the other hand, intramuscular injection of 11-KT significantly increased the level of motile sperm cells as well as the fertilization success [92]. Meanwhile, 12 different steroids were tested for their ability to induce final maturation in wolffish. Three out of twelve compounds were especially efficacious to induce final maturation. The three steroids were derivatives of progesterone with 17,20β-dihydroxy-4-pregnen-3-one (17,20β-P) as a naturally occurring compound in wolffish. This suggests that 17,20β-P might be a maturation-inducing compound. However, sulfation of 17,20β-P renders its maturation-inducing capacity ineffective, and the treatment of females with a high dose of 17,20β-P during the final maturation phase resulted in deleterious effects on egg quality and survival [Johnsen and Tveiten, unpublished results].

## 7. Gamete Collection, Quality, and Handling

In captivity spotted wolffish do not display normal spawning behavior and the females release unfertilized eggs [25,26]. For this reason, gametes need to be collected and fertilized artificially. Females need to be closely monitored to collect the oocytes within a short time window, just before they are released unfertilized into the water [1,15]. As the males produce sperm throughout the spawning season. After sedation, the gametes are collected and should be mixed in a way that maximizes the fertilization rate [26]. It appears, from preliminary experiences, that manual gametes collection in spotted wolffish can frequently result in excessive handling of the individuals and impacts males’ and females’ health. A small-scale study on female fitness suggests that females have a limited number of successful recoveries from manual egg extraction [Dupont-Cyr, unpublished results]. After 5–10 spawning seasons, females may develop uterine leiomyoma [Farley, unpublished results]. 

### 7.1. Oocytes

Wolffish are determinate multiple spawners and the females release all their eggs in a single yearly batch [5,93], and in captivity sometimes once every two years [1,93]. The descriptions of ovarian development for this species are scarce and only with wild captured individuals. It is believed that they present a group-synchronous ovarian development with single batches of eggs. Beese and Kändler [94] identified three generations of oocytes at the time of spawning: (1) the oocytes ready to be spawned (2) oocytes at the cortical alveolus stage that would constitute the next year’s batch and (3) primary oocytes. The vitellogenesis in the spotted wolffish is also protracted (5–6 months) like its close relative—the Atlantic wolffish [88] and is highly impacted by the temperature [87]. Gonadosomatic index (GSI) increases with age and shows pronounced peaks that lead to spawning. Moreover, the females have lower relative fecundity compared to other large-bodied teleosts [13]. Nonetheless, the fecundity increases with size [1,2,95]. Tveiten [96] describes the best age in terms of reproductive performance of spotted wolffish to be 9 years. In the wild, it has been observed that larger Atlantic wolffish females spawn earlier than small females [97]. Although no studies exist in captivity, personal records from Canada and Norway indicate that first-time spawners usually spawn later in the season and have smaller and low-quality eggs. Spotted wolffish egg quality was briefly evaluated by Tveiten et al. [98] and Desrosiers et al. [64] seeking to identify factors involved in egg viability.

#### 7.1.1. Collection and Handling

During the spawning season, females are easily recognized due to their large belly, which increases markedly a couple of days before the egg release [1,2,15]. This increase is probably related to the increased production of ovarian fluid [5]. Alternatively, ultrasound systems have been used to assess the degree of maturation by estimating the average diameter of the oocytes inside the oviduct [15]. Nonetheless, this method implies removing the individuals from the water and the associated stress. Due to the difficulties in tracking close spawning females, they are usually isolated in smaller tanks [15]. Then size increase in the genital pore opening marks the ovulation [1,2]. It is believed that the opening of the genital pore will allow copulation in the wild. However, in captivity, eggs can only be stripped when the opening reaches 5–7 mm, or when some eggs are released [5,26]. A mirror at the end of a stick is usually used to avoid disturbance of females (See Figure 4, Figure 5 and Figure 6) Other close-to spawning signs in captivity are usually observed, such as laying on the side, and have been described in detail elsewhere (e.g., cease feeding; teeth loss) [15]. Stripping prior to this stage frequently results in low-quality eggs [2]. Each female produces between 1 to 4 L of eggs (each L with 5000 to 6000 eggs, depending on the eggs’ size) [5], but some females can produce up to 30,000 eggs [2]. Figure 7 presents the relation between the number of eggs and the size of the females achieved over several years of monitoring at the MERINOV facilities. Detailed oocyte collection methodologies had been previously described [15,26].

Although there is no study looking at the effect of light on egg quality, it is believed that eggs are light-sensitive, and thus all authors unanimously recommended keeping the eggs in the dark and use of red filters during handling.

#### 7.1.2. Quality Evaluation

Due to the long egg incubation time (<5 months) and unpredictable egg quality, setting up accurate parameters for oocytes quality evaluation has been recognized as one of the main bottlenecks for the expansion of spotted wolffish farming [1,2,5,8]. The incubation of low-quality egg batches results in the labor-intensive work of removing dead eggs and increases the risk of the spread of diseases [2]. Parameters such as oocyte color, size and weight, and spawn volume are frequently recommended to record [15]. However, only the fertilization rate has so far been used as an indicator of which batches to keep or discard [1,2].

Similar to most fish species, the spotted wolffish fecundity increases with female size (See Figure 7). In the Atlantic wolffish, fecundity was lower when females were kept at 12 °C, compared with 4 or 8 °C, several months prior to spawning [87]. Nonetheless, no study has looked at the relationship between females’ fecundity and egg quality. As for egg color, reported colors are variable between spawns, from transparent to paler yellowish and strong yellow (almost orange) [91]. Nevertheless, as long as the egg batch presents a homogenous color, this parameter does not seem to be related to the quality of the eggs [Beirão et al., in preparation]. The spotted wolffish eggs are also characterized by the presence of multiple oil droplets [91], similar to other species with benthic eggs [99]. Regarding the egg size, reported values vary between 5 to 6.5 mm [13,91], and although bigger eggs result in heavier larvae, no correlation has been found with egg survival [35]. Whereas some works indicate that egg size increases with female age [13], other authors’ results support that there is no relation between egg size and female weight [1], but rather, first-time spawners have smaller eggs [91]. In the Atlantic wolffish, egg size is affected by the rearing temperature the fish are kept at in the months prior to spawning. Females kept at 8 °C have bigger eggs compared to those reared at 4 or 12 °C [87]. In most species, larger eggs have more metabolic reserves, thus producing larger larvae with higher chances of survival [100]. In the spotted wolffish, correlation between egg size and larvae weight was reported [35]. However, how exactly this parameter affects egg quality remains to be elucidated. Another important particularity is the ovarian fluid. Spotted wolffish oocytes are embedded in abundant ovarian fluid that constitutes up to 20–30% of the egg batch volume [25]. Some authors recommend the removal of the excess ovarian fluid, so it does not affect sperm-to-egg contact efficiency [15]. Nonetheless, the importance of ovarian fluid is not clearly established. Recent work indicates that low ovarian fluid pH is associated with low hatching [Beirão et al. in preparation]. Finally, overripe eggs are often obtained, especially toward the end of the spawning season [1], and most likely from egg bound females. Such events are presumably related to stress from excessive handling or advanced nephrocalcinosis that blocks the urogenital pores causing difficulties in egg release [Beirão, J., personal observation].

#### 7.1.3. Storage

After egg collection, sometimes artificial fertilization cannot be performed immediately due to sperm unavailability, and occasionally, the eggs needed to be transported between facilities. In the Atlantic wolffish, eggs stored in the ovarian fluid at 4 °C presented normal fertilization rates and cell cleavage even after 24 h [17,101]. No studies have look at egg storage of spotted wolffish. Nevertheless, routine techniques indicate that eggs can be stored in the ovarian fluid for up to 12 h, without quality degradation, as long as they are kept in the dark and refrigerated (4 °C) [15]. In some cases, eggs have also been fertilized after 24 h of storage with approximately 50% success in the fertilization rate [Beirão, J., personal observation].

### 7.2. Sperm

Males produce a small volume of semen somewhat disproportionate to their size. They have small testes and low GSI, which is stable around the year and does not increase with age [10]. Some authors have argued that the low sperm volume and concentration might be a captivity-induced dysfunction [5]. Males produce gametes throughout the spawning season [1], and in some cases even outside the spawning season [15]. However, the sperm collected outside the spawning season presents lower quality [Beirão, J., personal observation]. Sperm production outside of the spawning season might be used to attract females [2,15]. The low sperm concentration and volume and unusual motility is usually attributed to the presumed internal fertilization/insemination [5,25]. Spotted wolffish sperm is already motile upon stripping and inactivates when in contact with seawater or high osmolality [25,26]. The sperm is motile at osmolalities between 200 and 500 mOsm [25], indicative that it is delivered and swims within an environment of similar values, most likely the ovarian fluid. Similar observations regarding the osmolality at which the sperm swims were made in the marine sculpins with internal insemination but external fertilization [30].

#### 7.2.1. Collection and Handling

During the spawning season, males with good sperm quality can be stripped regularly without quality deterioration [1]. Usually, 2 to 4 weeks of resting time between stripping are recommended [102], but no study has looked at how multiple stripping frequency affects the sperm quality. Sperm collection methods for wolffish have been previously described in detail [26,103,104,105]. Typically, samples contaminated with metabolic wastes (usually yellowish in color or with crystal deposits) are discarded [26,105], as they either present very low motility or lose motility faster. Nevertheless, low motility sperm can still be utilized to fertilize the eggs if used immediately.

#### 7.2.2. Quality Evaluation

In the spotted wolffish, the most commonly used parameters include sperm volume, density, and various motility parameters [25,26,102]. Sperm volumes are rather low in this species, with reported values varying between authors. Individual volumes rarely exceed 2 mL [2], although some males can exceptionally produce 6 or 8 mL [1,2], or up to 14 mL after hormonal treatments [28]. However, reported values can be affected by the sperm collection method [26]. Sperm density is within the range of 10^9^ cells per mL [25,26] and is almost clear in appearance. Meanwhile, sperm concentration has been primarily evaluated using a haemocytometer [25,26,106], and other approaches such as spermatocrit or spectrophotometry have also been tested [26]. In the Atlantic wolffish, the spermatocrit value was lower when males were kept at 8 or 12 ˚C several months prior to the spawning season compared to 4 °C [87]. In the spotted wolffish, spermatocrit values ranged between 2–17% [26,28], which is a very low value compared to other marine species with values close to 90%: *Gadus morhua* [107]; *Hippoglossus hippoglossus* [108].

Sperm motility is undoubtedly the most frequently used sperm quality parameter. In the spotted wolffish, it has been assessed subjectively using a binocular microscope [15,109]. This method provides an initial assessment of the motility and is often used in fertilization trials [110]. Nonetheless, the use of computer-assisted sperm analysis systems (CASA) provides a more objective evaluation and has also been employed in spotted wolffish sperm quality studies [25,102,106,111]. The CASA systems, besides the percentage motility, measure different sperm swimming velocity parameters, such as curvilinear velocity (VCL) or straight-line velocity (VSL), parameters associated with the trajectory such as linearity (LIN), and other parameters such as wobble of the sperm head; for a detailed description, see [112]. Spotted wolffish sperm is characterized as slow-moving, with VCL and VSL values of 20–50 and 4–12 μm/s, respectively, and a circular trajectory (LIN = 24%) [25,102]. The slow sperm velocity in spotted wolffish (most species have initial sperm VCL > 100 μm/s [113], is probably linked with its long motility period (up to two days), in contrast with most species with motility lasting only for up to 2 min [113]. In addition, Kime and Tveiten [25] considered VCL and beat cross frequency (BCF) as the most sensitive movement indicators in the spotted wolffish. BCF is associated with the peculiar rapid side-to-side movement of the sperm head, wiggly behavior, and probably a swimming adaptation in the viscous ovarian fluid in which the eggs are embedded.

Other sperm quality parameters that have been evaluated in spotted wolffish include sperm morphology and seminal plasma. The mean head length of the sperm cells is 3.52 ± 0.58 µm [105]. On the other hand, the spermatozoa ultrastructure has never been studied in the spotted wolffish, but it is assumed to be similar to the Atlantic wolffish. In the Atlantic wolffish, the spermatozoa present a well-developed midpiece that contains numerous mitochondria [114]. The high number of mitochondria is typical of fish with internal fertilization which allows the sperm to stay motile for a prolonged period. Regarding the seminal plasma, the pH varies between 6.3–6.7 with osmolality values between 310–330 mOsm [28,106], but none of these seminal plasma parameters seem to be related to the sperm motility [106]. 

#### 7.2.3. Storage

It is often difficult to find spermiating males and with enough sperm volume at the end of the spawning season [1], and this represents a challenge to fertilize the last egg batches. Consequently, the development of sperm storage techniques has been extensively researched [25,102,105,106,109,111]. Cryobanking of the sperm of genotyped broodstock populations is justified in order to control undesirable inbreeding levels. Different studies have optimized different aspects of the cryopreservation process such as cryoprotectant and concentration [102,109], freezing and thawing rate [102,109], or cryopreservation volume [109,111]. Compared with other marine fish, spotted wolffish sperm is quite resilient to cryopreservation. Indeed, post-thawing survival of sperm to the cryopreservation process has been reported without the need to use diluents or cryoprotectants [105]. This resilience is probably linked to the presence of antifreeze proteins [105,115,116], which have been shown to improve sperm cryoresistance in other species [117]. Fine-tuning of the protocols has been performed using 0.5 mL straws [102]. Briefly, the samples are diluted in a 1:1 ratio with an extender [25] containing 10% dimethyl sulfoxide, loaded in 0.5 mL straws, and frozen gradually in liquid nitrogen at a freezing rate of −14.5 °C/min. However, 2–30 mL of cryopreserved sperm (depending on the sperm concentration) is needed to fertilize 1L of eggs (~5000 eggs) [26,102]. This suggests that the current procedure necessitates a substantial number of small straws which proved to be complicated and time-consuming. Thus, large cryovials (5–10 mL) are the option preferred by the industry [111]. 

As previously mentioned, the spotted wolffish sperm is already motile at stripping, and although it remains motile for a couple of days [25,106], it is currently not possible to deactivate and reactivate it. Under this scenario, refrigerated storage is only possible for the few days the sperm remains motile, and thus, sperm cryopreservation has become the best way to synchronize gametes availability. Nonetheless, the possibility to store spotted wolffish sperm by refrigeration has also been studied [106]. Compared to cryopreservation, this short-term storage protocol eliminates the complexity of the long-term storage which requires access to liquid nitrogen, reagents/chemicals, and trained personnel, which is difficult to carry out in small facilities. The results from Gonzalez-Lopez et al. [106] have shown that sperm samples diluted at 1:10 ratio in an extender with 1% bovine serum albumin (BSA), can be refrigerated at 2 °C for 1–2 days with the retention of high percentage motility.

## 8. Fertilization 

Due to its oviparous nature and absence of normal spawning behavior, both sexes need to be stripped, and fertilization must be performed artificially in vitro to ensure the production of larvae [5,8,26]. 

### 8.1. Gamete Mixture, Sperm: Egg Ratio and Contact Time

Since sperm in this species is already motile at stripping [25], dry fertilization in the ovarian fluid is the obvious choice used for the gamete mixture [1,91], as the addition of a fertilizing solution would dilute the gametes and affect their fertilizing capacity [26]. Because of the low sperm velocity, long gametes contact times (1 to 6 h) are required for this species in vitro fertilization [5,26] before water is added. This contact time is extremely long compared with most species, usually shorter than 30 min [118]. The duration of the gametes contact time is dependent on the sperm:egg ratio and the gametes need to be gently mixed every half an hour [1,2,5,26]. Gametes mixing, in nature, is probably obtained by the increased activity of the females, with something resembling convulsions, that are observed hours before the release of the eggs and after the initial genital opening [15]. Longer contact times will increase the success of sperm to encounter an egg. When sufficient sperm of good quality is available (at least 5 × 10^5^ sperm cells per egg) fertilizations close to 100% can be obtained with just 1 h contact time [26]. Alternatively, low sperm volume available in the range of thousand sperm cells per egg can be compensated by giving up to 6 h contact time (Figure 8). However, extremely long contact times also imply increased labor and there might be a decrease in egg quality. 

Most authors recommend the use of sperm from 2–3 males to fertilize the batch of eggs, mainly to avoid negative effects of low sperm quality from one male or to avoid possible male–female incompatibilities. Nonetheless, how exactly this could lead to sperm competition and affect genetic diversity is not known [119]. Because of the lower sperm quality observed after cryopreservation, specific fertilization protocols have also been developed. In this case, the decrease in sperm quality can be compensated either by increased contact time or by increasing the volume of sperm used. Current protocols recommend a ratio of at least 5 × 10^5^ sperm:egg for a contact time of 2 h, or 5 × 10^4^ if the contact time increases (4 or 6 h) [102].

### 8.2. Egg Stickiness

When the inseminated egg batch is introduced in the water, the eggs become stuck to each other, and it is extremely difficult to individually remove eggs without disturbing the eggs surrounding them. Nonetheless, contamination with microorganisms or simple unfertilized or aborted eggs that start to deteriorate need to be removed [35] (see Section 9). While some authors recommend the eggs to be stirred for a couple of hours to prevent them from getting stuck together, or just spread enough [2,120], few have tested different treatments in an attempt to remove the eggs’ stickiness and facilitate individual egg removal during incubation and increase the efficiency of disinfection protocols. Removal of ovarian fluid is also sometimes recommended for other fish species displaying egg adhesiveness either by washing the eggs with an isosmotic solution or with the addition of clay, but to our knowledge, this was never tested for wolffish. Nonetheless, high survival is obtained when the ovarian fluid is not removed and the eggs are allowed to sit and stick together in double or triple layers, as long as the initial quality of the eggs is good [1]. Batches with a low percentage of fertilization or that present contamination should be discarded due to the extra labor that this implies.

## 9. Egg Incubation

Spotted wolffish egg incubation ranges between 800 and 1000 day-degrees (DD) [1] and the species embryonic development has been described elsewhere [91]. The extended period of incubation lasting for several months (at a temperature of 6 °C this corresponds to more than 5 months) proves to be space-consuming and labor-intensive [2]. Dead and decaying eggs need to be removed with some frequency before they serve as a substrate for microorganisms which could represent a significant amount of labor in poor quality egg batches [1]. Nonetheless, good egg batches could present survival and hatching rates close to 90%. The uncertainty in egg survival represents one of the most important constraints in this species’ commercial production. Mortality during incubation is generally higher during the first third of the incubation that lasts until the eye-stage or 300 DD [35,121] and relatively low after that. Although it is reported some mortality also occurs during the hatching period [120,122]. Thus, technology that secures the highest possible survival during this sensitive stage is urgent.

Egg fertilization success, easily evaluated at 2–4 cell stages between 12 and 24 h (depending on the incubation temperature) post-fertilization, is recommended to evaluate in order to decide whether or not to proceed with incubation or to discard low fertilization batches [91]. Nonetheless, Hansen and Falk-Petersen [35] did not observe a relation between fertilization rate and egg survival. Early cell cleavage quality has also been suggested as a parameter to evaluate the egg batches’ quality at an early stage of the incubation stages [91].

### 9.1. Incubation Environment

Egg incubation temperature has been shown to affect egg survival and development, day-degrees until hatching, larvae survival, and growth after hatching [35,121]. In the Atlantic wolffish, high incubation temperatures have been linked to structural damages in skeletal development [93]. Different temperatures have been tested in the spotted wolffish, ranging between 2 and 8 °C [35,120,121], inclusively decreasing temperatures [1]. The optimal incubation temperature, based on egg survival, hatching rate, and larvae quality, is usually recommended to be 6 °C [35,91,121]. According to the latter, eggs incubated at a higher temperature of 8 °C hatch with more DD (up to 1200 DD) and require fewer incubation days than eggs incubated at 4 °C, which can hatch after 750 to 800 DD. Lower temperatures, such as 4 °C, result in larger and heavier larvae in a more advanced stage of development and with smaller yolk sacks; however, higher mortalities are observed during the egg incubation. Simultaneously, decreasing temperatures from 8 °C at the beginning to 3 °C at hatching also gave positive results. However, there is a strong maternal effect on both egg survival to different incubation temperatures, time of hatching, and larvae quality [121]. As for larvae survival after hatching, this was observed to be higher for larvae from eggs incubated at 6 °C compared with larvae from eggs incubated at 8 and 4 °C [35]. Rapid temperature changes can also be the cause of egg mortality, and in the later stages will cause early hatching. Similarly, exposure of eggs to high or medium-light intensities resulted in elevated mortalities [120]. 

### 9.2. Disinfection

Due to the long egg incubation period, eggs are usually sensitive to micro-organisms infection [123], mainly bacteria such as *Flexibacter* sp., but also protozoan, as identified in the Atlantic wolffish [104]. These micro-organism infections lead to the eggshell ulceration, and have been associated with premature hatching after 400 DD in Atlantic wolffish [101,104]. Indeed, the complement component C3 part of the innate immunity has been already detected in unfertilized eggs, and it is believed to be an important part of the defensive factors of developing embryos against bacterial infections [124]. Egg batches with high levels of infections should be discarded because of the high amount of extra work that they imply. Even so, disinfection routines are required in good egg batches to control and prevent mortalities caused by microorganisms [91,123]. These disinfection routines are usually accompanied by routines to remove dead eggs once a week. The higher the frequency of the egg disinfection, the higher the likelihood to cause early hatching. Therefore, depending on the level of appearance of micro-organisms infestation, the disinfection should be performed on a regular basis, either once per month or bi-weekly with 150 ppm of glutaraldehyde for 5 min [123]. Other disinfectants are currently used, such as iodine-based disinfectants. Excessive disinfection or high concentrations of disinfectant can lead to a hardening of the eggshell that results in a failure to hatch [123]. Egg disinfection should also be avoided or reduced after 600 DD, although if needed, disinfections can be conducted after this period. Additionally, the closer this procedure is conducted to 900 DD the higher the likelihood that it triggers early hatching [Beirao, J., personal observation]. Current work should focus on reducing the number of chemicals needed for disinfections and increase the water treatment techniques [2].

### 9.3. Incubation Technology and Egg Transportation

Upwelling systems, usually 20 L, are the preferred vessels used by commercial wolffish farms in order to incubate newly hatched spotted wolffish [121], Figure 9. The continuous water flow of 2.5 to 3 L·min^−1^ that secures high enough levels of oxygen is used in Norwegian and Canadian facilities. Low oxygen can cause early hatching [120]. Initial attempts to incubate wolffish eggs in Canada used incubation systems similar to salmonids (vertical incubators with local upwelling in individual incubation drawers) [15,122]. However, these systems present the disadvantage of the spread of contamination between batches in different trays and potentially insufficient oxygen supply. The drawers and water inlet with slight modifications ensure optimal water mixing and oxygenation. 

Mechanical disturbances can also affect eggs’ survival [120,121]. Nonetheless, after the eyed stage, approximately 300 DD, and until 600 DD eggs can be transported relatively safely with low mortalities. This transport could last for a few hours, but a few days of transport are also possible given sufficient water, oxygenation, and cooling precautions. 

### 9.4. Hatching and Newly Hatched

At around 900–1000 DD, hatching, if not spontaneously, is often mechanically induced, for example by gently shaking the incubators [35], or by increasing the water temperature in 1–2 °C, or by gentle squeezing of the eggs using cheesecloth followed by gentle release back in the incubators. The relatively small yolk sack is believed to be used primarily for energy rather than tissue synthesis [121]. The hatching process in this species is relatively energy-demanding since the yolk conversion efficiency of unhatched embryos of similar age has been observed to always be higher. The larvae length and weight at hatching increase parallel with the hatching DD [91]. Premature hatching has been observed from 400 DD [1,91]. However above 600 DD, mechanical stress, such as excessive handling, often leads to premature hatching [120,121]. This premature hatching, when it occurs before 750–800 DD, is usually followed by the mortality of the hatched larvae [1]. On the other hand, it is frequent that a significant number of well-developed embryos fail to hatch [35]. This phenomenon is attributed to the hardening of the eggshell, preventing the larvae from hatching at the required 900–1000 DD at 5–6 °C. In addition, post-mature hatching can also occur, which can lead to the absence of feeding behavior, and accounts for a significant number of dying embryos or low larvae survival [120]. Savoie et al. [122,125] proposed that failure to feed post-hatching at 900–1000 DD could be related to egg quality (incubation protocols, broodstock nutrition) but could be improved significantly by proposing protein hydrolysate. The exact reasons remain to be elucidated. 

Spotted wolffish hatch as well-developed and robust, at approximately 20–25 mm [91], with a small yolk sack, and a fully functional digestive system. Newly hatched spotted wolffish are usually transferred rapidly and cultured in raceway systems or correctly sized circular or Swedish tanks displaying low water levels and slow circular current conditions to facilitate feed intake of floating particles (Figure 10). Newly hatched wolffish require a surface rather than volume as they are not active swimmers. They usually stand on the bottom resting on the pelvic fins. Strand et al. [126] proposed the concept of stacking low-level water raceways for Atlantic wolfish cultivation that is also adapted for spotted wolffish. No live feed (rotifers, artemia) is needed, as they can be fed directly on a dry diet [5,126]. Growth is best at a temperature gradually raised after transfer from 4–5 °C to 8–10 °C. Lamarre et al. [40,41] and many other authors locate optimal temperatures for growth between 8 and 12 °C, but common practices in the industry all suggest temperatures between 8 and 10 °C during the early rearing phase to minimize mortality that can occur when approaching higher temperatures. Once the critical stage of first feeding is completed, mortality is generally quite low thereafter. Further information on the early life stages cultivation practices can be found in Falk-Petersen et al. [1], Foss et al. [2], and Le François et al. [5,8]. 

## 10. Future Research and Conclusions

The four major areas of research that should be tackled for the improvement of the reproductive performance of spotted wolffish broodstock and also contribute to the achievement of continuous and predictable production of high-quality eggs and larvae are:(1)Development of Specific Diets

At this stage of its development, no commercial-specific diet adapted to spotted wolffish broodstock is available. The industry relies on formulated feed for marine fish. Floating characteristics and low lipid content are required to ensure, respectively, optimal feed intake at high rearing densities and control of fatty liver disease. Nephrocalcinosis caused by unbalanced mineral content could also be controlled by identifying optimal mineral composition in the formulated diets; 

(2)Health and Welfare

General lack of knowledge of the reproductive biology of this species has been an impediment for the implementation of the best practices for broodstock handling and management. Future studies should focus on stress reduction and improvement in the health and reproductive performance of broodstock fish;

(3)Establishment of Breeding Programs

No breeding program exists at this point. The complete absence of selective breeding programs represents a hindrance for new candidate species aquaculture [127,128]. Nonetheless, the scarce supply of spotted wolffish eggs, together with the late maturation experienced by this species, causes major constraints by the few companies involved in farming spotted wolffish at this stage;

(4)Improvement of Egg Incubation and Early-Stage Cultivation Techniques

Improvement of egg incubation is without room for discussion; it is the most labor-intensive stage in spotted wolffish production and is still the center of high level of uncertainties. Future work should focus on spawner selection, egg incubation technology, and first feeding technology that could improve survival.

## Figures and Tables

**Figure 1 animals-11-02849-f001:**
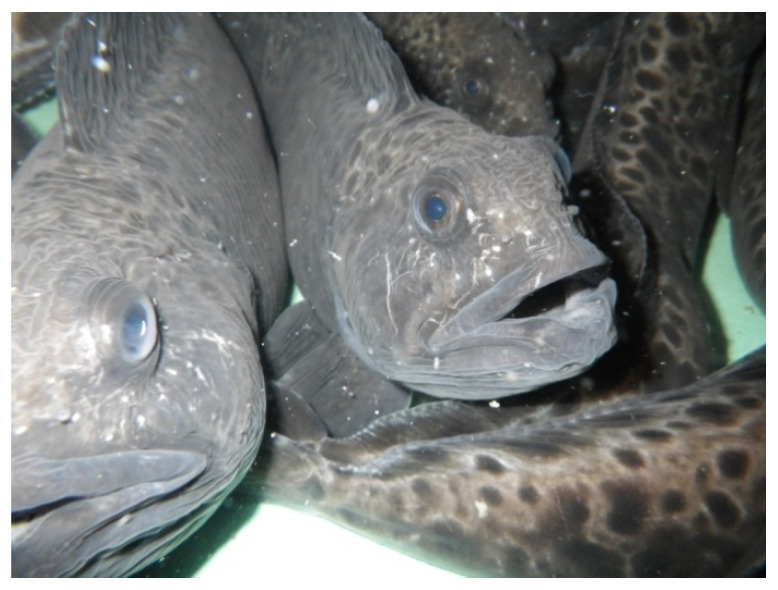
Subadults spotted wolffish in on-growing tanks at Maurice Lamontagne facilities (DFO/MPO, Mont-Joli, QC, Canada). Source: UQAR, Savoie, A.

**Figure 2 animals-11-02849-f002:**
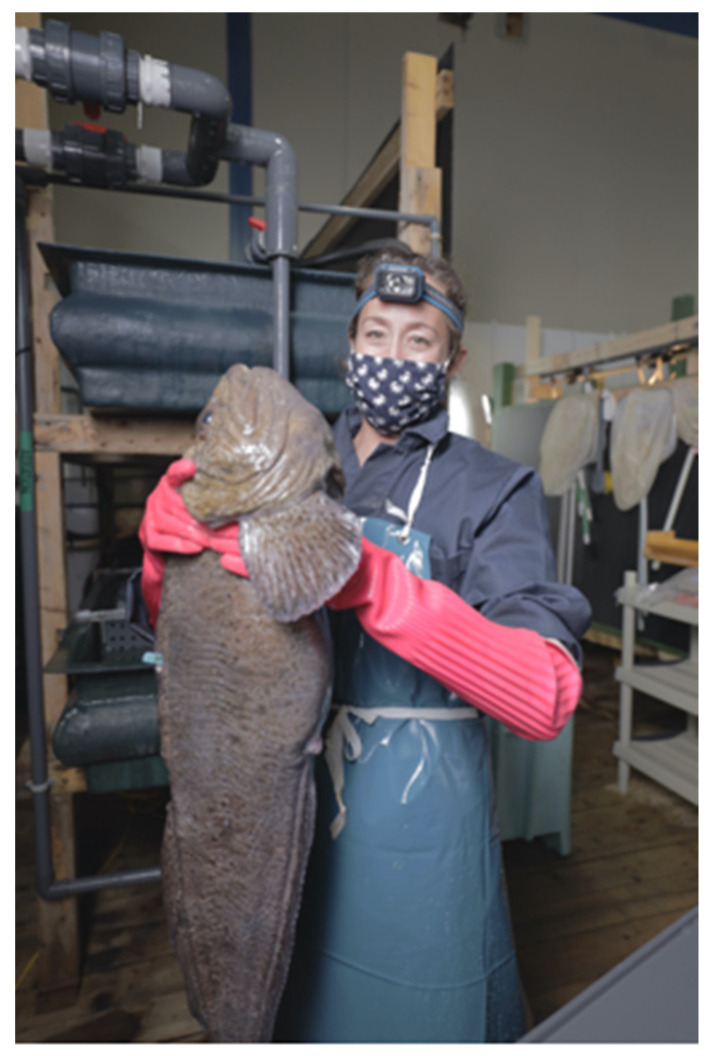
Broodstock fish in preparation for examination (blood sampling, semen or egg extraction or measurements) at MERINOV facilities (Grande-Rivière, QC, Canada). Source: MERINOV, Dupont-Cyr, B.-A.

**Figure 3 animals-11-02849-f003:**
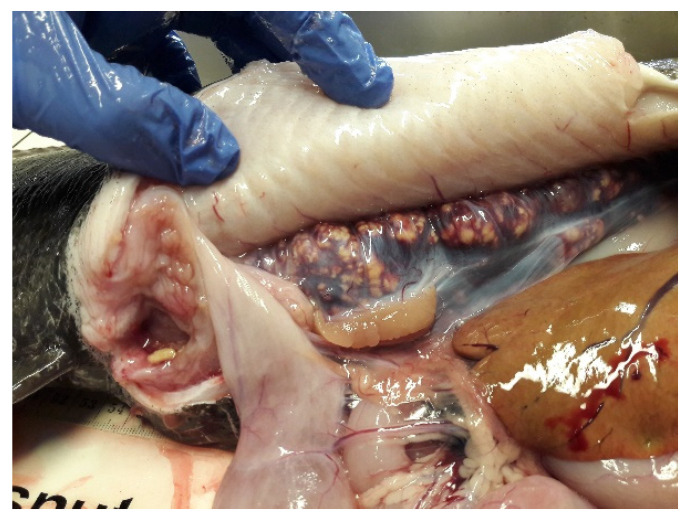
Extreme example of nephrocalcinosis with calcareous deposition all along the kidneys of a mature farmed-origin male sacrificed in 2018 at Nord University (Bodo, Norway). Source (J. Beirão).

**Figure 4 animals-11-02849-f004:**
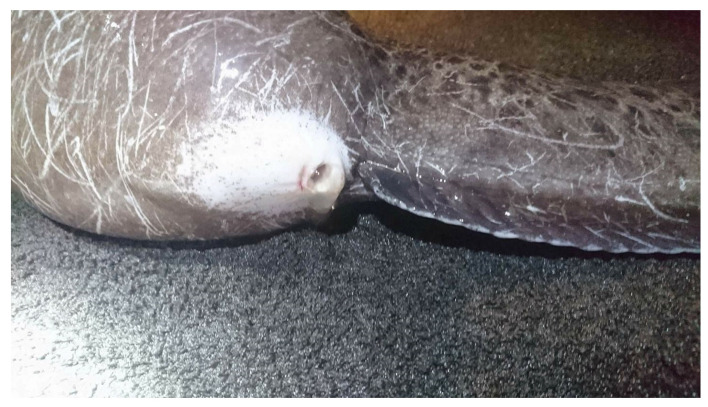
Ventral view of a female with a wide genital opening (>5 mm) and observable release of ovarian fluid. The large round belly is also clearly visible. Picture obtained before manual collection of the eggs. Source: Beirao, J.

**Figure 5 animals-11-02849-f005:**
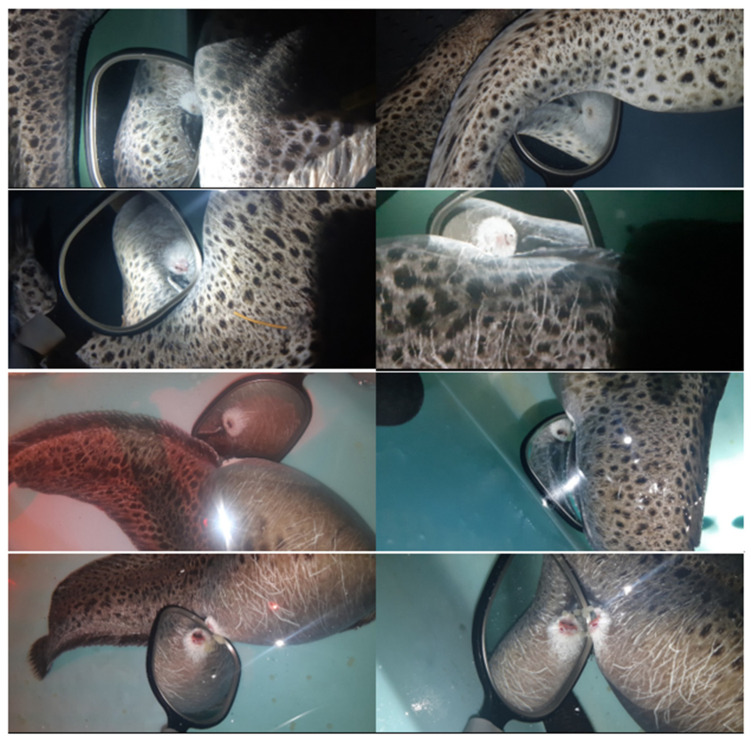
Use of a mirror to monitor genital pore opening of the females preparing to spawn. Source: MERINOV, Dupont-Cyr, B.-A.

**Figure 6 animals-11-02849-f006:**
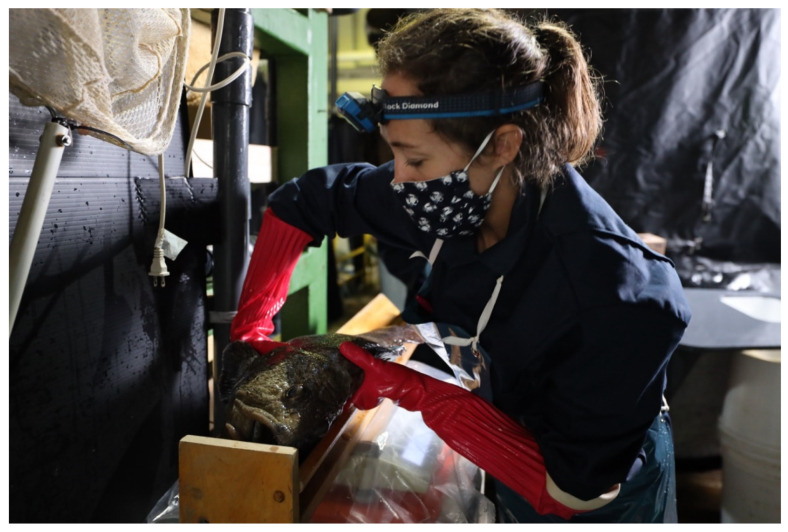
Handling of broodstock fish at Merinov facilities (Grande-Rivière, QC, Canada). Source: MERINOV, Dupont-Cyr, B.-A.

**Figure 7 animals-11-02849-f007:**
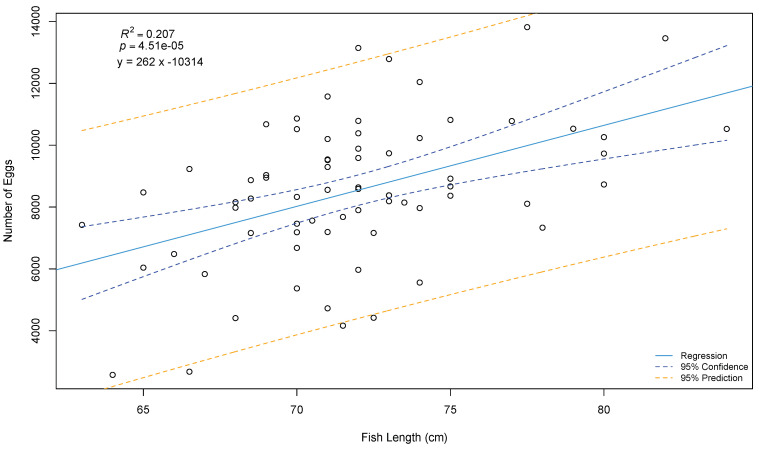
Regression of egg number over fish length (cm) recorded from 1999–2020 (unpublished data). Source: Merinov, Dupont-Cyr, B.-A.

**Figure 8 animals-11-02849-f008:**
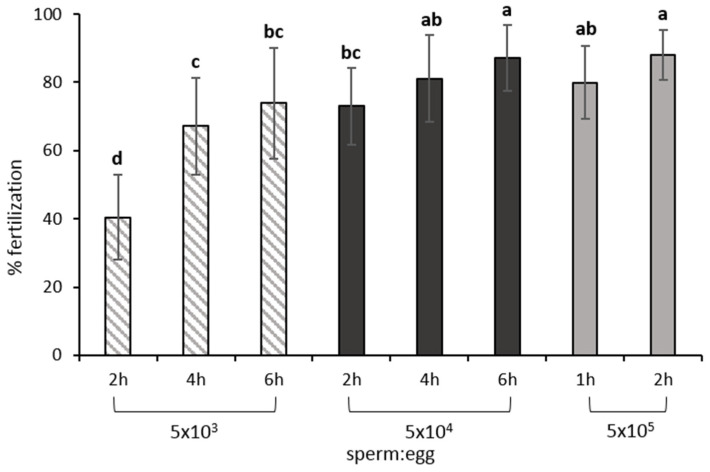
Percentage of fertilized eggs in spotted wolfish relative to the contact time and the sperm to egg ratio used. Bars with stripes for 5 × 10^3^ sperm per egg, dark gray bars for 5 × 10^4^ sperm per egg and light gray bars for 5 × 10^5^ sperm per egg. Different letters stand for significant differences detected with Duncan’s multiple comparison test (*p* < 0.05). The error bars represent the standard error of the mean (*n* = 4). Adapted from Beirão and Ottensen [26].

**Figure 9 animals-11-02849-f009:**
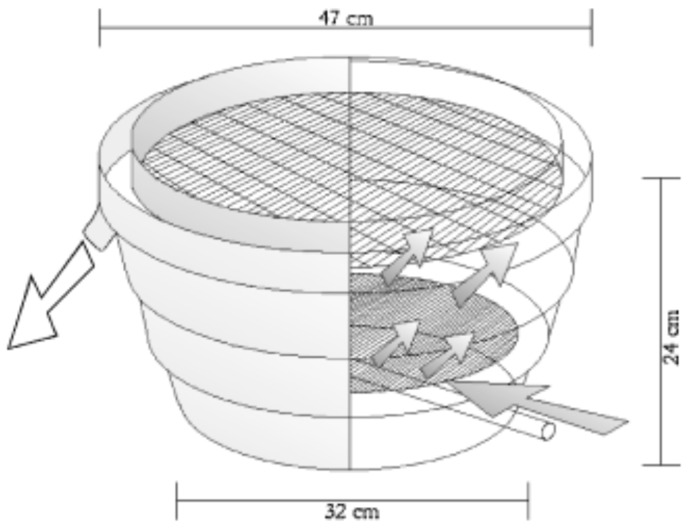
Schematic figure of the 20 L incubators used for spotted wolffish eggs incubation. Eggs are deposited in the superior tray. The gray and white arrows represent water inlet and outlet, respectively. Adapted from Sund and Falk-Petersen [121].

**Figure 10 animals-11-02849-f010:**
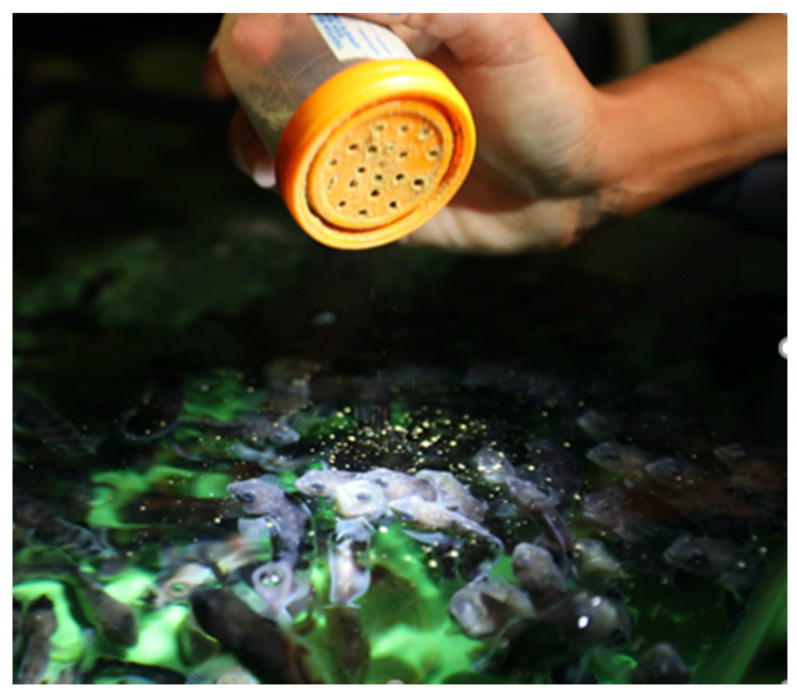
Feeding of juveniles in rearing tank. Source: Merinov, Dupont-Cyr, B.-A.

## Data Availability

Not applicable.

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
