# Peer review of "Spotted Wolffish Broodstock Management and Egg Production: Retrospective, Current Status, and Research Priorities"

_animals, 2021, doi:10.3390/ani11102849_

Round 1
Reviewer 1 Report
Dear Authors,
I found your review very useful for specialists in aquaculture and those fish biologists who are interested in the spotted wolffish and other species of the genus, the structure of the paper leaves much to be desired.
Taking into account the entire volume of work done by you, I ask you to make one more effort and take into account the comments that were made to the text of the article.
Please pay attention to the links to sources in the text. I didn't find 34, 75, 96, 107.
Please change the order of the links to the figures. In the text, they go like this: 1, 2, 3, 7, 8, 4. Others are not mentioned, although you give ten figures.
I would like Figure 5 to consist of one or two photos but of a larger size; Figures 8 and 9 to have a better resolution.
Pay attention to how the rules of the Journal regulate the syntaxis: long dashes for periods, ranges, intervals; cent degree signs, comas for tens of thousands, super- and subscripts.
The paper is assigned as a review, but there are sections, which have 0 references (please see the file).
I would like you to think of the recombining of Section 2, upward the features of the sexual dimorphism and, probably, making an independent subsection within this section.
And the main thing that I lacked in this work was actualization. It seemed to me that you are reasoning as if it was an attempt to justify yourself for the whole amount of work you have done. Please make the article meaningful. Link aquaculture and fishing, maintaining populations, etc.
With all my respect
Author Response
Please pay attention to the links to sources in the text. I didn't find 34, 75, 96, 107.
Reply: 34 is at page 18, 75 page 14, 107 at page 18 and 96 (that was missing is now at page 18).
Please change the order of the links to the figures. In the text, they go like this: 1, 2, 3, 7, 8, 4. Others are not mentioned, although you give ten figures.
Reply: We checked the ms, and found that the figures are all in order. However, only figure 10 was not mentionned, now it is, and Figure 9 was incorrectly labeled Figure 4. It was corrected. Here at the Figures location in the text.
Figure 1 and 2: page 3
Figure 3: page 9
Figure 4,5,6 and 7: page 14
Figure 8: page 20
Figure 9: page 22
Figure 10: page 23
Pay attention to how the rules of the Journal regulate the syntaxis: long dashes for periods, ranges, intervals; cent degree signs, comas for tens of thousands, super- and subscripts.
Reply: we did the necessary revision of the entire ms and applied corrections where needed. Ex. Headings with capital letter at the beginning of the words, long dashes, degrees, Kg was modified to kg, mL to ml, um to µm
I would like you to think of the recombining of Section 2, upward the features of the sexual dimorphism and, probably, making an independent subsection within this section.
Reply: we changed the subsection in section 2 as recommended.
2.1 Reproductive strategy page 3
2.2 Sexual dimorphism, page 4
2.3 Internal fertilization/insemination page 4
I would like Figure 5 to consist of one or two photos but of a larger size; Figures 8 and 9 to have a better resolution.
Reply: Figure 8 was inserted as a .tif file. We asked to the editorial office of Aquaculture Research for a better quality file for figure 9 (an original version or so). We will need to wait if that is a critical issue. As for figure 5, we wanted to display the variability in what can be observed during spawning season at the urogenital region of female fish. We think it is interesting as is since there is a close-up at Figure 4.
The paper is assigned as a review, but there are sections, which have 0 references (please see the file).
Reply: The sections that lack references are our understanding, as experts in the field of wolffish cultivation, of the general features of wolffish cultivation and status of the aquaculture industry aimed at wolffish (section 1. Spotted Wolffish Farming) or rely on pers. observations and analysis (Section 3) most of the information is hands-on experience and the work is not published. We believe, that we used this hands-on knowledge sharing reasonably without references. Specific sections dealing with the domestication traits are all covered with references further in the appropriate sections.
And the main thing that I lacked in this work was actualization. It seemed to me that you are reasoning as if it was an attempt to justify yourself for the whole amount of work you have done. Please make the article meaningful. Link aquaculture and fishing, maintaining populations, etc.
Reply: as the reviewer might acknowledge, wolffish aquaculture research is associated with a small number of researchers committed to it. The essential work has been conducted by a few of us over the last 20 years or so. It is difficult to extract ourselves from the founding work we collectively conducted. The literature cited is updated to the fullest and meaningful studies are cited with no regard to who or when. It is a paper on aquaculture/cultivation not fisheries as the title suggests. We have added information in relation to fisheries when appropriate and used over 128 references including the newest publication focusing on reproduction and aquaculture, 34 of which we are, logically, co-authors since we are heavily involved in applied research of wolffishes. We have added information in relation to fisheries when appropriate.
Reviewer 2 Report
This is a well-written review on a topic on which there is no existing review from earlier. The emphasis is on reproduction on the species in question, as this is considered the major obstacle in aquaculture. However, the review also covers general aquaculture information of the spotted wolffish, as well as the limited work that has been carried out on diseases. As cultivation of the spotted wolffish is now in progress in two countries, it can be assumed that the review will be of great use to students and scientists working on these projects, although their numbers may still be quite limited. The review may also be of interest to those studying fish reproductive biology in aquaculture in more general terms.
Author Response
We thank reviewer #2. We included information that related to all aspects of reproduction in the wolffish from capture or broodstock fish to fertilization all the way to first feeding of newly-hatched. Disease and stress response were also covered briefly as these research area will gain more importance as the production levels increase.
Round 2
Reviewer 1 Report
Dear Authors,
numerous misprints remained in the manuscript (esp. supersripts, degree signs). I have highlighted some of them in the file. Please review the entire text, not just the fragments I have highlighted, again.
The semantic part of the text currently suits me completely. Thanks for the corrections and clarifications.

Author Response
The necessary corrections were applied. At the exception of the use of power of ten, I don't really understand what is wrong with the usual use of 5 × 105 and your suggestions are not very clear.
